# A Contention-Based Hop-By-Hop Bidirectional Congestion Control Algorithm for Ad-Hoc Networks

**DOI:** 10.3390/s19163484

**Published:** 2019-08-09

**Authors:** Jiashuai Wang, Xiaoping Yang, Ying Liu, Zhihong Qian

**Affiliations:** College of Communication Engineering, Jilin University, Changchun 130012, China

**Keywords:** ad-hoc network, MAC, congestion control, contention window, hop-by-hop, bidirectional

## Abstract

Existing hop-by-hop congestion control algorithms are mainly divided into two categories: those improving the sending rate and those suppressing the receiving rate. However, these congestion control algorithms have problems with validity and limitations. It is likely that the network will be paralyzed due to the unreasonable method of mitigating congestion. In this paper, we present a contention-based hop-by-hop bidirectional congestion control algorithm (HBCC). This algorithm uses the congestion detection method with queue length as a parameter. By detecting the queue length of the current node and the next hop node, the congestion conditions can be divided into the following four categories: 0–0, 0–1, 1–0, 1–1 (0 means no congestion, 1 means congestion). When at least one of the two nodes is congested, the HBCC algorithm adaptively adjusts the contention window of the current node, which can change the priority of the current node to access the channel. In this way, the buffer queue length of the congested node is reduced. When the congestion condition is 1–1, the hop-by-hop priority congestion control (HPCC) method proposed in this paper is used. This algorithm adaptively changes the adjustment degree of the current node competition window and improves the priority of congestion processing of the next hop node. The NS2 simulation shows that by using the HBCC algorithm, when compared with distributed coordination function (DCF) without congestion control, the proposed unidirectional congestion control algorithms hop-by-hop receiving-based congestion control (HRCC) and hop-by-hop sending-based congestion control (HSCC), and the existing congestion control algorithm congestion alleviation—MAC (CA-MAC), the average saturation throughput increased by approximately 90%, 62%, 12%, and 62%, respectively, and the buffer overflow loss ratio reduced by approximately 80%, 79%, 44%, and 79%.

## 1. Introduction

With the rapid development of the Internet of Things, the application of ad-hoc networks is becoming increasingly widespread [1]. There will be more and more sensors accessing the network to transmit data. A large amount of data aggregation will inevitably lead to network congestion, and network congestion will bring about multiple network problems such as packet loss, network delay, throughput reduction, and excessive energy consumption [2]. Therefore, it is necessary to study congestion control in ad-hoc networks [3]. Wan et al. [4] proposed an end-to-end congestion control algorithm called Congestion Detection and Avoidance (CODA). The node determines whether congestion occurs according to the buffer occupancy ratio and the channel load. When congestion occurs, congestion can be alleviated by adjusting the sending rate of the source node to reduce the receiving rate of the congested node. However, end-to-end congestion control has a strong dependence on round-trip time, which inevitably leads to packet loss [5]. In contrast, the hop-by-hop congestion control protocol has a faster response speed. Hull et al. [6] pointed out that the end-to-end congestion control algorithm cannot satisfy the congestion control of ad-hoc networks in multi-hop environments because of the higher distribution of ad-hoc networks. Feng et al. [7] proposed a distributed adaptive algorithm with hop-by-hop congestion information feedback and proved that the hop-by-hop congestion control algorithm is superior to the end-to-end algorithm in queue length. The network layer is based on the end-to-end transmission control protocol (TCP), so in order to achieve hop-by-hop congestion control, it must pass the medium access control (MAC) layer [8].

Various papers [9,10,11,12,13,14,15,16,17,18,19,20] have proposed hop-by-hop MAC layer congestion control algorithms. In the present paper, the hop-by-hop algorithm for increasing the sending rate is called sending-based congestion control, and the algorithm for suppressing node receiving rate is called receiving-based congestion control. Papers [9,10,11] involve sending-based congestion control, and papers [11,12,13,14] involve receiving-based congestion control. Wang [9] proposed an upstream hop-by-hop congestion control (UHCC) protocol based on cross-layer design. It took advantage of unoccupied buffer size and the traffic rate of each node as congestion level indications, based on which every upstream traffic rate is adjusted with its node priority to mitigate congestion hop-by-hop. Kim [10] suggested a transport-controlled MAC protocol (TC-MAC) that combines the transport protocol into the MAC protocol. TC-MAC provides a fairness-aware lightweight transport control mechanism. When congestion occurs, the congested node continuously sends data to achieve network congestion mitigation. Basaran et al. [11] proposed a lightweight distributed congestion control method. Each node detects congestion by considering the queue lengths and channel conditions. According to the estimated congestion degree, each node dynamically adjusts the data packet transmission rate. Qian et al. [12] proposed a novel MAC protocol to alleviate the congestion—CA-MAC (congestion alleviation—media access control), which allows the nodes with more buffered packets to transmit with a higher probability, as well as an intelligent burst packet transmission when the congested nodes seize the channel. Yi et al. [13] developed a fair hop-by-hop congestion control algorithm. By restricting the access time of upstream nodes, the rate of receiving data from congested nodes can be reduced to alleviate congestion. Jang [14] proposed a congestion control technique for duty cycling wireless sensor network MAC protocols (CCDC). It detects congestion by checking the current queue size. If it detects congestion, it suppresses the node receiving rate by adding supplementary wake-up times. Sadeghi et al. [15] presented a simple and efficient hop-by-hop layer 2 congestion control scheme (SECC), which utilizes local information in the nodes to detect congestion and compute the target rate for the congested flows. SECC utilizes unicast signaling to inform the upstream nodes of the congestion status. The upstream nodes then modify their transmission rate to the target rate specified by the congested downstream node.

However, the above congestion control algorithms only control the congestion of a single node. When congestion occurs at both adjacent nodes in one hop, congestion mitigation at the current node may aggravate congestion at the next hop. In a tree-like network, the next hop node often acts as the forwarding hub of multiple paths. Once the congestion alleviation of one path causes the congestion of the central node, it will inevitably lead to the paralysis of other paths. Moreover, congestion mitigation speed is limited by the fact that there is only one direction of congestion control. Papers [16,17] proposed new congestion control algorithms to improve throughput but did not solve the problem of packet loss caused by congestion. Papers [18,19] proposed congestion control algorithms for cross-layer design, which is more complicated than operating only through the MAC layer. Paper [20] is a congestion prevention method, which controls the media access according to the priority of node messages and reduces conflicts by using different random competition windows. When congestion occurs, the algorithm cannot achieve the effect of congestion mitigation.

Aiming at the above problems, this paper proposes a new contention-based hop-by-hop bidirectional congestion control algorithm—HBCC. By detecting the queue length of the current node and the next hop node, the congestion conditions can be divided into the following four categories: 0–0, 0–1, 1–0, 1–1. For the three conditions of congestion, the algorithm proposes three new congestion control strategies—hop-by-hop receiving-based congestion control (HRCC), hop-by-hop sending-based congestion control (HSCC), and hop-by-hop priority congestion control (HPCC). In this way, not only can congestion be minimized, but also the central node, which plays a more important role in the network, can have a higher priority in congestion processing. Ad-hoc is composed of a large number of sensor nodes. It is a multi-hop self-organizing network system formed by wireless communication of sensor nodes. These sensor nodes are organized into clusters and they communicate with the cluster heads using IEEE 802.15.4 [21]. The cluster heads report the harvested data to the control center using IEEE 802.11. This paper will verify the performance of the HBCC algorithm in an IEEE 802.11 context.

The remainder of this paper is organized as follows: in Section 2, we discuss the problems of existing protocols. In Section 3, we present the main idea of our algorithm and its details. Meanwhile, in Section 4, we discuss the simulation results of our algorithm and prove that it has better performance in average saturation throughput and buffer overflow loss ratio. Finally, we conclude the paper in Section 5.

## 2. Problems of Existing Network Protocols in Congestion Control

The existing hop-by-hop congestion control algorithms are unidirectional, but unidirectional congestion control algorithms have two problems: effectiveness and limitations. In order to describe these two problems more clearly, we take a simple network topology as an example to illustrate the two problems.

### 2.1. Network Topology

Ad-hoc networks are distributed sensing networks with the sensor that senses and examines the outside world. The sensor nodes distributed in the detection area send the received data to the aggregation node through the intermediate node. Due to this characteristic of ad-hoc networks, we can represent the local ad-hoc network with a simple tree topology, which is shown in Figure 1. In Figure 1, node0, node1 and node2 are located at the end of the tree topology as sensor nodes for data acquisition. Then, node3 and node4 are the intermediate nodes to forward the collected data to the sink node5. The other parameters of the network topology are set as shown in Table 1. We model each wireless link between any two nodes in the network to have a finite positive capacity. Each data flow in the network corresponds to an ordered sequence of links. In this wireless system, a single transmission is intended for only one receiver, and each node has only a single transceiver, and hence only half-duplex communication is allowed. Further, a node can successfully receive from at most one other node at any given time.

### 2.2. Effectiveness Problem

The idea of the unidirectional congestion control algorithm is that when a node is congested, according to the congestion condition, the node adjusts the data transmission or reception rate to control the network load and achieve the purpose of congestion mitigation. However, when both nodes of the adjacent hop are congested, the unidirectional control algorithm will not work. In the network topology of Figure 1, when both node3 and node4 are congested, if the data transmission rate of node3 is increased to alleviate the congestion of node3, a large number of data packets are sent to node4. However, at this time, node4 is also congested, so the data packet sent by node3 due to the mitigation of congestion is discarded, which is a serious problem for the network. Otherwise, the method of reducing the data transmission rate of node3 is used to alleviate the congestion of node4. Because node3 is congested, the reduction in data transmission rate will increase the congestion level. This is the effectiveness problem in the unidirectional control algorithm.

### 2.3. Limitations Problem

A unidirectional control algorithm does not consider the congestion state of the next hop node in the network. In the topology shown in Figure 1, since node3 has three child nodes, the fact that node0 is waiting to access the channel will inevitably result in the accumulation of data packets. As the number of packets is increased, node0 will experience congestion. Node3 has three data input sources, but only one output, so the node3 buffer queue will accumulate some data packets. If the data transmission rate is increased in order to alleviate the congestion of node0, then node3 will inevitably experience the accumulation of data packets, and congestion will occur. The congestion of node3 causes node1 and node2 to fail to forward data through node3. Reducing congestion at node0 will make node1 and node2 unable to send data, which is detrimental to the entire network. Unidirectional congestion control methods have not considered that increasing the transmission rate of congested nodes will obstruct the transmission of other nodes, which is the limitation of unidirectional congestion control algorithms.

## 3. HBCC Algorithm

In order to solve the problems raised in Section 2, this paper proposes a contention-based hop-by-hop bidirectional congestion control algorithm—HBCC. The algorithm proposes bidirectional congestion control to avoid the effectiveness problem and limitations of unidirectional congestion control. By detecting the queue length of the current node and the next hop node, the congestion conditions can be divided into the following four categories: 0–0, 0–1, 1–0, 1–1. According to the four congestion conditions, this paper proposes several new congestion mitigation strategies—HRCC, HSCC, and HBCC—which relieve the congestion the most. The meaning of bidirectionality is that the congestion information of two adjacent nodes is taken into account when congestion control is carried out. That is to say, if a node in the network is congested, the algorithm should consider the congestion information of the last hop and the next hop in the link before congestion mitigation. The node congestion control of HBCC is not only based on the node’s own congestion situation, but also on the congestion situations of the last hop and the next hop in relation to this node. We call this algorithm a bidirectional congestion control algorithm, which considers the congestion information of upstream and downstream nodes.

### 3.1. Congestion Detection

Congestion detection methods can be divided into two categories: detection based on either queue length or channel information [22]. The idea of the congestion detection method based on channel information is to calculate the channel conditions (such as channel load, data service time, control frame transmission frequency, etc.) over a period of time. According to these channel conditions, the node judges whether congestion is occurring or not. The idea of the congestion detection method based on queue length is to set a buffer queue threshold. When the current buffer queue length is greater than the threshold, it is judged that the node is congested, otherwise it is considered that no congestion is occurring. It is obvious that the congestion detection method based on queue length can more quickly reflect the congestion condition of the current node [23,24]. Therefore, this paper selects the detection method based on queue length to detect the congestion of nodes.

In the HBCC algorithm, the current queue length *q* is compared with the set congestion threshold *k*, and when *q* is greater than *k*, it is judged that congestion is occurring in the network. Through several simulations under the network topology shown in Figure 1, it is found that when *k* is 0.75 Q_max_, the effect of congestion control is the best. Therefore, the value of the congestion threshold *k* is taken as 0.75 Q_max_, and Q_max_ is the maximum queue length of the node.

### 3.2. Congestion Notification

In the MAC layer protocol, the data transmission process is as shown in Figure 2. After the node accesses the channel, the data transmission mode is as shown in Figure 3.

When congestion is detected, neighboring nodes should be informed so that they can adopt the correct measure against the generated congestion. Congestion information can be propagated explicitly or implicitly. Some congestion control protocols notify about the congestion by setting a congestion notification (CN) bit in the packet header [22]. Unlike the explicit method, implicit congestion notification does not insert further load to the network and the congested nodes. In this type of congestion notification scheme, the congested nodes inform other sensor nodes by piggybacking the congestion information in a payload packet header. A number of congestion control protocols apply acknowledge character (ACK) signaling to indicate the congestion state [9,25,26,27]. 

As can be seen from Figure 2, when the node is sending successfully, it will initialize the competition window (*CW*). As shown in Figure 3, after the receiving node receives a data packet, it will feed back an ACK control frame to the sending node. In this paper, a congestion flag *CI* is set in the ACK returned by the receiving node, which records the congestion status of ACK sending nodes. The congestion control algorithm in this paper is bidirectional. Therefore, two congestion flags need to be set at the current node: one is *Temp_CI* to save the local congestion condition, the other is *Next_CI* to save the next hop congestion condition. The value of *Next_CI* is taken from the value of *CI* in the received ACK. The modified ACK frame format is shown in Figure 4. The procedure of congestion condition transmission is shown in Algorithm 1. According to the values of these two congestion flags, the congestion conditions of the two nodes are divided into four categories: 0–0, 0–1, 1–0, 1–1. The classification is shown in Figure 5.

**Algorithm 1**: Pseudo code for the congestion condition transmission algorithm

**Input:**
  temp_queuelength:  Buffer queue length of current node.  Queue_max:     Maximum buffer queue length of nodes.  *k*:              Congestion threshold.  queuelength:        Used to get the length of the buffer queue.
**Output:**
  *CI*:           Next hop congestion marker in ACK.  next_queuelength:    Next hop buffer queue length in ACK  *Next_CI*:       The variable to save the congestion condition of next hop in current node.  *qnext*:             The variable to save the length of buffer queue of next hop in current node.**Initially:**  *CI* ← Null  next_queuelength ← Null    *Next_CI* ← Null   *qnext* ← Null**if** node needs to send an ACK **then**  queuelength ← temp_queuelength  **if** 0 ≤ queuelength ≤ *k*
**then**    *CI* ← 0  **else if**
*k* < queuelength ≤ Queue_max **then**    *CI* ← 1    next_queuelength ← queuelength  **end if****else if** node received an ACK **then**  *Next_CI* ← *CI*  *qnext* ← next_queuelength
**end if**



### 3.3. Congestion Control

Congestion control is classified into the following four categories: (1) Traffic control [28]: in this technique, congestion is mitigated by means of reducing the number of packets injected into ad-hoc networks. (2) Resource control [24]: in this case, congestion is handled by, for example, increasing network resources or using other idle or uncongested paths for the transmission of data towards the sink. (3) Priority-aware congestion control scheme [9]: in this scheme, congestion is managed by considering different priorities in congestion situations. (4) Queue-assisted technique [23]: congestion is tackled by the queue length of the nodes. Our proposed congestion control method (HBCC) combines (1), (3) and (4). In HBCC, the queue length of nodes is used as the symbol of congestion control, and then the queue length of nodes is reduced as much as possible by using the method of traffic control. HBCC includes two traffic control methods: HRCC and HSCC. If the effect of traffic control is not good, the priority-aware congestion control scheme is used. The priority-aware congestion control scheme in HBCC is HPCC. The congestion control of HBCC is shown in Figure 6.

#### 3.3.1. DCF Algorithm

DCF is used to transmit data in the 802.11 protocol, and DCF provides a standard competitive service like Ethernet. DCF is the basic access control mode of the IEEE802.11 protocol. In DCF mode, after detecting the busy channel, the node uses the CSMA/CA mechanism and random backoff time to share the wireless channel. The basic rule of DCF is that the node listens to the busy condition of the surrounding medium through the carrier sense mechanism. If the channel is busy, the sending node will continue to monitor the channel. If the channel is idle, in order to avoid collision between nodes, the nodes enter a random backoff state. The generation of backoff time is as follows:(1)$$Backoff_{Time}=Random()\times SlotTime$$

Here, *Random* () is a random number between (0, *CW*).

#### 3.3.2. Hop-By-Hop Receiving-Based Congestion Control—HRCC

When *Next_CI* = 1, the unidirectional congestion control algorithm to suppress the reception rate of the next hop node should be adopted. According to this principle, the hop-by-hop receiving-based congestion control (HRCC) algorithm is proposed in this paper. The idea of the HRCC algorithm in this paper is: the current node detects that the next hop congestion flag *Next_CI* = 1, that is, the next hop node is congested. By increasing the *CW* (competition window) of the current node, the priority of the current node to access the channel is reduced, so the data sending rate of the current node is reduced [29]. That is, the data receiving rate of the next hop node is reduced, so that the congestion of the next hop node is alleviated.

It can be seen from Equation (1) that the larger the value selected by *Random* (), the longer the backoff time before the current node sends the data. As the waiting time of the current node increases, the receiving rate of the next hop node gradually decreases. Therefore, when the next hop node is congested, the next hop node queue length is used as a parameter to adaptively adjust the size of the contention window of the current node, thereby achieving the purpose of congestion mitigation. The *CW* of the current node is adjusted in the following way:(2)$$CW=\left(CW_{\min}+1\right)\times 2^{n}-1$$

Here, *CW* is the adjusted initial contention window of the current node, CW_min_ is the contention window minimum, and *n* is the window adjustment parameter, which is given by:(3)$$n=\left(\left(q_{next}-k\right)/\left(Q_{\max}-k\right)\right)\times \log_{2}\left(\left({\mathrm{CW}}_{\max}+1\right)/\left({\mathrm{CW}}_{\min}+1\right)\right)$$
where *q*_next_ is the queue length of the next hop node, *k* is the congestion threshold, Q_max_ is the maximum length of the node buffer queue, and CW_max_ is the maximum contention window.

According to Equations (2) and (3), the adjustment range of the *CW* is between CW_min_ and CW_max_. When *q*_next_ = *k*, *n* = 0 and *CW* = CW_min_. As the length of the node queue increases, the *CW* also increases, and the priority of the node to access the channel is continuously decreasing. When *q*_next_ = Q_max_, *n* is given by:(4)$$n=\log_{2}\left(\left({\mathrm{CW}}_{\max}+1\right)/\left({\mathrm{CW}}_{\min}+1\right)\right)$$

By bringing Equation (4) into Equation (2), the value of *CW* is obtained as CW_max_, at which point the current node gets the lowest priority to access the channel. The current node waits for the longest time, and the next hop node will have enough time to send the packets in the buffer queue.

#### 3.3.3. Hop-By-Hop Sending-Based Congestion Control—HSCC

When *Temp_CI* = 1, the unidirectional congestion control algorithm to improve the sending rate of the current node should be adopted. According to this principle, the hop-by-hop sending-based congestion control algorithm (HSCC) is proposed in this paper. The idea of the HSCC algorithm in this paper is: when *Temp_CI* = 1, the priority of the current node to access the channel is increased by reducing the *CW* of the current node, so that the data packets in the buffer queue of the current node are sent out as soon as possible.

It can be seen from Equation (1) that the smaller the value selected by the number of *Random* (), the shorter the backoff time before the current node sends the data. The higher the probability of the current node accessing the channel, the higher the data transmission rate of the current node. Therefore, when the current node is congested, the HSCC algorithm uses the current node queue length as a parameter to adaptively adjust the size of the *CW* of the current node. *CW* is given by:(5)$$CW=\left({\mathrm{CW}}_{\min}+1\right)*2^{m}-1$$
where *m* is given by:(6)$$m=-\left(\left(q_{temp}-k\right)/\left(Q_{\max}-k\right)\right)$$

Here, *q*_temp_ is the queue length of the current node. It can be seen from Equation (6) that when *q*_temp_ = *k*, *m* = 0, so *CW* = CW_min_. As the current node queue increases, *m* is reduced, *CW* is reduced, and the priority of the current node to access the channel is continuously increased. When *q*_temp_ = Q_max_, *m* = −1 and *CW* = 0.5 × (CW_min_ − 1). At this time, the current node obtains the highest priority to access the channel, and the current node can send the data packets of the buffer queue more quickly, so that the congestion is alleviated.

#### 3.3.4. Hop-By-Hop Priority Congestion Control—HPCC

The above two methods of congestion control only consider the congestion conditions of one node, and they adjust the data packet transmission rate of the current node by changing the priority of access to the channel according to the length of the node buffer queue, thereby achieving the purpose of relieving congestion. However, unidirectional congestion control algorithms have the problem of validity when two nodes are congested at the same time (see description in Section 2). In order to solve this problem, this paper proposes the Hop-by-hop Priority Congestion Control (HPCC) algorithm. When the current node congestion flag *Temp_CI* and the next hop node congestion control *Next_CI* are both 1, in order to ensure that the next hop node has a higher congestion processing priority, the node uses HPCC to adjust the *CW* of the current node, which is given by:(7)$$CW=\left\{ \begin{array}{l} CW_{\mathrm{next}}\\ {\mathrm{CW}}_{\min} \end{array}\right.\begin{array}{cc} & q_{\mathrm{temp}}\geq q_{\mathrm{next}}\\ & q_{\mathrm{temp}}<q_{\mathrm{next}} \end{array}$$

Here, *CW*_next_ is the congestion contention window of the next hop node, *q*_temp_ is the current node buffer queue length, and *q*_next_ is the next hop node buffer queue length. It can be seen from Equation (7) that if *q*_temp_ ≥ *q*_next_, the algorithm sets the *CW* of the current node to be equal to the *CW* of the next hop node. Due to the fact that the current node has a greater degree of congestion but the next hop node has a higher congestion handling priority, the same congestion processing level is set. If *q*_temp_ < *q*_next_, the algorithm sets the *CW* of the current node to CW_min_. This is because the buffer queue of the next hop node is longer, and the *CW* setting of the next hop node is less than CW_min_, which also ensures that the next hop node has a higher congestion processing priority. The HBCC algorithm not only ensures that the congestion of the current node is alleviated, but it also ensures that the next hop node has a higher congestion processing priority.

#### 3.3.5. Contention-Based Hop-By-Hop Bidirectional Congestion Control—HBCC

HBCC deploys a hop-by-hop approach for congestion control. There are three phases of the HBCC protocol, i.e., congestion detection, congestion notification and congestion control. In the congestion detection phase, the HBCC protocol uses buffer occupancy. It uses the congestion threshold value to calculate the level of congestion at each node. If the queue length is higher than the threshold, the child nodes need to decrease their data rates to avoid congestion and packet loss. Changing the value of *CW* can indirectly change the data rate of nodes. HBCC provides higher network throughput by reducing packet loss.

The proposed HBCC algorithm combines the four congestion processing algorithms. When the local congestion sign *Temp_CI* and the next hop congestion sign *Next_CI* are both 0, neither node has congestion, and so the DCF algorithm is adopted. When *Temp_CI* = 1 and *Next_CI* = 0, the current node is congested and the next hop node is not congested. Therefore, it is most suitable to improve the current node transmission rate to alleviate congestion. At this time, the proposed HSCC algorithm is used to adjust the competition window. When *Temp_CI* = 0 and *Next_CI* = 1, the current node is not congested and the next hop node is congested. Therefore, the best way to alleviate congestion is to suppress the next hop receiving rate. At this time, the proposed HRCC algorithm is used to adjust the competition window. When *Temp_CI* = 1 and *Next_CI* = 1, that is, congestion occurs at both nodes, the HPCC algorithm proposed in this paper is used to adjust the competition window. The HBCC algorithm considers the congestion conditions of two nodes and adopts different contention window adjustment methods for different congestion conditions.

As shown in Figure 1, if node3 is congested and node4 is not congested (1–0 congestion condition), we adopt the HSCC algorithm for congestion control. The contention window of node3 is adjusted using Equation (5) to obtain a higher priority to access the channel, but the priority of node4 is not improved, which is only in order to transfer the data packets buffered by node3 to node4. Although the packet loss ratio at node3 can be reduced, a transfer delay may be caused due to the fact that the transferred data packet cannot be transmitted immediately. If both node3 and node4 are congested, node4 should not only alleviate local congestion but should also forward packets sent by node3 as soon as possible. Therefore, the time for node3 to use high priority should be reduced in order to reduce the chances of competing with node4 for the channel. In order to solve this problem, we have added an adaptive data transmission rate adjustment mechanism to increase the packet transmission rate in order to reduce its priority usage time when the congestion node transmits data. *R*_data_ is given by:(8)$$R_{data}=\left\{ \begin{array}{l} 10\;\mathrm{Mbps}\\ 5\;\mathrm{Mbps}\\ 2\;\mathrm{Mbps}\\ 1\;\mathrm{Mbps} \end{array}\right.\begin{array}{cc} & \left(q_{temp}-q_{next}\right)\geq 0.25Q_{\max}\\ & 0.1Q_{\max}\leq \left(q_{temp}-q_{next}\right)<0.25Q_{\max}\\ & 0\leq \left(q_{temp}-q_{next}\right)<0.1Q_{\max}\\ & else \end{array}$$

It can be seen from Equation (8) that the larger the difference between the queue length of the current node and the next hop node, the larger the data sending rate of the current node. It is obvious that the larger the buffer queue length of the node, the more serious the congestion of the node; therefore, *CW* is adjusted to be smaller and the priority of the node is adjusted to be higher. Nodes with high priority also have high data transmission rates, which reduces the time that high priority nodes occupy the channel. Therefore, the transferred data packet can be transmitted as soon as possible. Through several simulations under the network topology shown in Figure 1, we find that when the segment threshold is set to 0.25Q_max_ and 0.1Q_max_, the congestion control effect is the best, so we adopt 0.25Q_max_ and 0.1Q_max_ as the segment threshold. However, this optimization of Equation (8) does not take into account the quality of the channel when improving the transmission rate, which may lead to the loss of data packets due to a poor channel. So this optimization method is only suitable for the case of good channel quality. Different delay optimization methods can be selected in different situations, but this does not affect HBCC congestion control to alleviate network congestion. The HBCC algorithm is shown in Algorithm 2.

**Algorithm 2**: Pseudo code for the HBCC algorithm

**Input:**
  temp_queuelength: Buffer queue length of current node.  Queue_max:     Maximum buffer queue length of nodes.  *k*:          Congestion threshold.  *Next_CI*:      The variable to save the congestion condition of next hop in current node.  qnext:       The variable to save the length of buffer queue of next hop node in current node.  queuelength:    Used to get the length of the buffer queue.  *Temp_CI*:     The variable to save the congestion condition of current node.  CW_min_:        The minimum contention window.
**Output:**
  *CW*:        Adjusted competition window of current node.  datarate:      Adjusted data sending rate of current node.  CWnext:      The variable to save the current CW value of the next hop node in current node.**Initially:**        *CW* ← CW_min_    datarate ← 1 Mbps**if** node needs to send a data **then**  queuelength ← temp_queuelength  **if** 0 ≤ queuelength ≤ *k*
**then**    *Temp_CI* ← 0  **else if**
*k* < queuelength ≤ Queue_max **then**    *Temp_CI* ← 1  **end if**  **if**
*Temp_CI* = 0 and *Next_CI* = 0 **then**    *CW* ← CW_min_  **else if**
*Temp_CI* = 0 and *Next_CI* = 1 **then**    *CW* is calculated by (2)  **else if**
*Temp_CI* = 1 and *Next_CI* = 0 **then**    *CW* is calculated by (5)  **else if**
*Temp_CI* = 1 and *Next_CI* = 1 **then**    CWnext is calculated by (5) with next_queuelength as parameter.    **if** temp_queuelength > next_queuelength **then**      *CW* ← CWnext    **else**      *CW* ← CW_min_    **end if**  **end if**  **if**
*Temp_CI* = 1 **then**    defferent ← temp_queuelength − next_queuelength    **if** 0 ≤ defferent < 0.1 * Queue_max **then**      datarate ← 2 Mbps    **else if** 0.1 * Queue_max ≤ defferent < 0.25 * Queue_max **then**      datarate ← 5 Mbps    **else if** defferent ≥ 0.25 * Queue_max **then**      datarate ← 10 Mbps    **end if**  **end if**
**end if**



#### 3.3.6. Complexity Analysis

Time complexity: The overall time complexity is computed as follows:(9)$$\begin{array}{ll} \Rightarrow TC & =O\left(n\right)+O\left(1\right)+O\left(1\right)\\ & \Rightarrow TC=O\left(n\right) \end{array}$$

Space complexity: The space complexity comes out to be as shown below:(10)$$\begin{array}{ll} \Rightarrow SC & =O\left(n\right)+O\left(1\right)+O\left(1\right)\\ & \Rightarrow SC=O\left(n\right) \end{array}$$

## 4. Simulation and Analysis

In this section, we first analyze the effectiveness of the algorithm in the topology shown in Figure 1, and then use the aggregation network topology and random network topology to analyze the algorithm robustness and related network performance.

### 4.1. Performance Metrics

The performance metrics used for evaluation in this paper are as follows:(1)Throughput [30]: This is defined as the number of packets received by the destination per unit time. A higher throughput reflects a higher efficiency of the network and is desirable.(2)Average end-to-end delay: This refers to the time taken for a packet to be transmitted from the source to its destination. The average end-to-end delay is the average end-to-end delay for all Throughput the packets that are successfully delivered. A lower delay reflects a higher efficiency of the network.(3)Buffer overflow packet loss ratio [31]: This is the ratio of packet loss caused by buffer overflow to the total number of packets sent to the network. A lower loss ratio reflects the reliability of the protocol.

### 4.2. Simple Tree Topology

#### 4.2.1. Parameter Settings

The simple tree network topology is shown in Figure 1, and the related parameter settings are shown in Table 2. After 10 seconds of simulation, node0, node1 and node2 start generating CBR packets, which are forwarded by node3 and node4 and transmitted to node5. Node1 and node2 generate data packets at a fixed rate of 10 packets/s, gradually changing the packet transmission rate of node0 (10 packets/s to 90 packets/s), which makes the network change gradually from no-congestion to congestion. Network performance is tested under the DCF, HBCC, HSCC, HRCC and CA-MAC [11] congestion control algorithms.

#### 4.2.2. Simulation Results and Algorithm Effectiveness Analysis

(1) Throughput

Figure 7 shows the curve of the average saturated throughput of node5 with the DCF, HBCC, HSCC, CA-MAC and HRCC algorithms, varying with load, under a simple tree topology. As can be seen from Figure 7, when the network load is small, the throughput of the four protocols increases linearly with the increase in input load. As the network load continues to increase, the network gradually becomes congested, and the throughput stops growing, in turn, in the DCF algorithm (when the load is 25 packets/s), the HRCC algorithm (when the load is 15 packets/s), the HSCC algorithm (when the load is 35 packets/s), the CA-MAC algorithm (when the load is 15packets/s) and the HBCC algorithm (when the load is 40 packets/s). The average saturation throughputs of the DCF, HRCC, HSCC, CA-MAC and HBCC algorithms are approximately 29 packets/s, 34 packets/s, 49 packets/s, 34 packets/s and 55 packets/s, respectively.

It can be seen from the simulation results in Figure 7 that the throughput performance of HBCC using hop-by-hop bidirectional congestion control is better than the HSCC algorithm, HRCC algorithm, CA-MAC algorithm and DCF algorithm. The average saturation throughput in HBCC is about 12% higher than in the HSCC algorithm, about 62% higher than in the HRCC algorithm, about 62% higher than in the CA-MAC algorithm, and about 90% higher than in the DCF algorithm. Because HBCC adopts bidirectional congestion control, when a node is congested, the node can handle the congestion quickly according to the congestion condition judged, so that the packet of the congested node can be sent out as soon as possible. The congestion is reduced, and the waiting time of the buffered data packet in the queue is also reduced, so that the data packet is sent to the destination node5 as soon as possible. Therefore, the throughput of the network is greatly improved.

(2) Buffer overflow packet loss ratio

Figure 8 shows the curves of buffer overflow loss ratio with network load using the DCF, HBCC, HSCC, CA-MAC and HRCC algorithms, under a simple tree topology. As can be seen from Figure 8, the buffer overflow packet loss ratio of the four algorithms increases with increasing network load. It can be seen from the figure that when the DCF algorithm has a load of 10 packets/s, the HRCC algorithm has a load of 15 packets/s, the HSCC algorithm has a load of 25 packets/s, CA-MAC algorithm has a load of 15 packets/s, or the HBCC algorithm has a load of 30 packet/s, the network begins to lose data packets. As the load on the network increases, the buffer overflow packet loss ratio of the HBCC algorithm is lower than that of the other three algorithms.

It can be seen from the simulation results in Figure 8 that the buffer overflow packet loss ratio performance of HBCC with bidirectional congestion control is better than the HSCC algorithm, HRCC algorithm, CA-MAC algorithm and DCF algorithm. When the load is 40 packets/s, the buffer overflow packet loss ratio using the HBCC algorithm is about 44% lower than the HSCC algorithm, about 79% lower than the HRCC algorithm, about 79% lower than the CA-MAC algorithm, and about 80% lower than the DCF algorithm. Because HBCC adopts bidirectional congestion control, when a node is congested, it can adaptively adjust the *CW* of the previous hop node to obtain the lower priority of access to the channel and suppress the data receiving rate of the current node. It can also adaptively adjust the contention window of the current node to obtain a higher priority of access to the channel and improve the sending rate of the current node. In this way, the congested node can reduce the length of the buffer queue of the current node more quickly. Therefore, the congestion of the network is alleviated, and the buffer overflow packet is avoided.

(3) Average end-to-end delay

Figure 9 shows the curves of average end-to-end delay with network load using the DCF, HBCC, HSCC, CA-MAC and HRCC algorithms, under a simple tree topology. As can be seen from Figure 9, the average end-to-end delay of the four algorithms increases with increasing network load. It can be seen from the figure that the average end-to-end delay of the four algorithms is, from high to low: HRCC, DCF, HBCC, HSCC.

Because the idea of the HRCC algorithm is to increase the time for the last hop node to wait for access to the channel in exchange for congestion mitigation, this will inevitably lead to a certain delay increase, but in exchange for a lower packet loss ratio and higher throughput. This is worthwhile for the entire network, because packet loss ratio is the most important network performance evaluation index. The average end-to-end delay of the HSCC algorithm is the lowest, because when the node is congested, by increasing the priority of access for the congested node, the time that the node waits for access to the channel is reduced, and the time of the data packet in the buffer queue is reduced. So, the end-to-end delay is lower. However, the CA-MAC algorithm does not have the delay optimization added in this paper, so the delay is still very large compared with HRCC. The HBCC algorithm combines two algorithms, so the average end-to-end delay curve lies between the two algorithms, and the congestion mitigation effect is better. Through the analysis of the above three network performance indicators, we can see that the HBCC algorithm can greatly improve the average saturation throughput of the network, reduce the average end-to-end delay, and greatly reduce the network buffer overflow packet ratio.

(4) Analysis of the congestion control process of the HBCC algorithm

Here, we analyze how the HBCC algorithm works by tracking the length of the buffer queue. Figure 10 shows the distribution of the buffer queues of node0, node1 (the buffer queue of node2 is similar to that of node1), node3, and node4 in the simulation time when the network loads are 20 packets/s and 25 packets/s. Figure 11 shows the distribution of the buffer queues of node0, node1, node3 and node4 in the simulation time when the network loads are 30 packets/s and 35 packets/s. In Figure 10 and Figure 11, the green histogram is the DCF algorithm, the red histogram is the HBCC algorithm, the brown is the intersection areas of two algorithms, the horizontal axis is the length of the buffer queue, and the vertical axis is the number of occurrences.

As can be seen from Figure 8, when the network load is 20 packets/s, the DCF algorithm has generated packet loss, and the HBCC algorithm has not generated packet loss. As shown in Figure 10c, the buffer queue length of node3 in the DCF algorithm has reached the full queue, while the buffer queue of node3 in the HBCC algorithm has not reached the full queue. When the network load is increased to 25 packets/s, the full queue has appeared at node0 under the DCF algorithm due to the larger quantity of data packets generated. The length of the buffer queue of node3 continues to increase. However, under the HBCC algorithm, the buffer queues of node0 and node3 do not reach the full queue. When the load is heavier, the packet loss ratio of the DCF algorithm is aggravated, while that of the HBCC algorithm does not occur. These results are also confirmed in the packet loss ratio curve of Figure 8.

From the change in queue length, we can see that network congestion has been greatly alleviated. This is because when it is detected that the buffer queue is larger than the congestion threshold, the contention window is adaptively adjusted according to the queue length, and the adjusted contention window changes the node data transmission rate. The congestion condition of node1 and node3 is 0–1, and the *CW* is adjusted using Equation (2). The buffer queue length of node1 is increased, and the data reception rate of node3 is reduced. It can be seen from Figure 10b that the buffer queue of node1 is slightly increased, and the buffer queue of node3 is below the full queue. The congestion condition of node3 and node4 is 1–0. At this time, the *CW* is adjusted by Equation (5). The adjusted *CW* increases the data transmission rate of node3, and a certain data packet is sent to node4. It can be seen from Figure 10d that the buffer queue of node4 increases. Figure 10c shows that the buffer queue of node3 is reduced to below the full queue. When the load is 25 packets/s, the congestion condition of node0 and node3 is 1–1, and the *CW* is adjusted according to Equation (7). Node3 preferentially handles congestion, and the buffer queue of node3 falls under the HBCC algorithm, so the congestion of node0 can also be relieved. Figure 10a shows that the buffer queue of node0 is also reduced.

The queues analyzed in Figure 10 are all cases where HBCC does not experience congestion. It can be seen from Figure 8 that the HBCC algorithm has generated congestion when the load reaches 35 packets/s. Therefore, according to Figure 11, this paper analyses the change in nodes’ buffer queue length from non-congestion to congestion. As shown in Figure 11, under the DCF algorithm, as the load increases, the buffer queue lengths of node0 are increasing, and a large number of them are clustered near the full queue. The buffer queue of node3 is saturated (the delay is also confirmed by the delay curve shown in Figure 9, but it can be seen from Figure 8 that the network packet loss ratio is increasing. The buffer queue lengths of node1 and node4 are very low. Under the HBCC algorithm, the buffer queues of node0 and node3 are greatly reduced, while the buffer queue lengths of node1 and node4 are both increased.

Comparing the changes in the buffer queues of nodes under the two protocols, it can be seen that when the congestion is aggravated, the congestion mitigation strength of the HBCC algorithm is gradually increased, and the length of buffer queues of congested nodes decreases to a great extent. This is because under the HBCC algorithm, when the length of the buffer queue increases, the adaptive adjustment of the competition window becomes very strong. Under the DCF algorithm, when the network loads are 30 packets/s and 35 packets/s, node0 has reached the full queue, and the buffer queue length of node3 is also clustered around 50. However, the buffer queue lengths of node1 and node4 are very low. Under the HBCC algorithm, the congestion condition of node0 and node3 is 1–1. The HBCC algorithm adjusts the *CW* of node0 according to Equation (7) and changes its data sending rate. After the congestion of node3 is alleviated, node0 also experiences an alleviation of congestion to a certain extent. As can be seen from Figure 11a, the buffer queue of node0 is basically below the full queue. Under the HBCC algorithm, the congestion condition of node1 and node3 is 0–1. The HBCC algorithm adaptively adjusts the *CW* of node1 according to Equation (2). The adjusted contention window reduces the data transmission rate of node1. The buffer queue of node 1 shown in Figure 11b has been greatly improved, which reduces the data receiving rate of node3. The congestion condition of node3 and node4 is 1–0. The HBCC algorithm adjusts the *CW* of node3 according to Equation (5) to increase its data sending rate. As can be seen from Figure 11c, the buffer queue length of node3 is basically below the full queue. The length of the buffer queue in node 4 has increased, as shown in Figure 11d.

The node queue status under the two algorithms at different loads is shown in Table 3. The adjustment effect of the HBCC algorithm is shown in Table 4. Table 4 shows that when the load is 20 packets/s, the average buffer queue length of each node in the HBCC algorithm decreases compared with the DCF algorithm. Because the degree of congestion is low at this time, the nodes in the HBCC algorithm are mostly in a no-congestion state. Therefore, compared with the DCF algorithm, the length of the buffer queue in non-congested nodes is basically unchanged. When the load is 25 packets/s, the average buffer queue length of node0 in the HBCC algorithm is higher than that in the DCF algorithm, but there is no full queue in node0, and no packet loss occurs. Other buffer queue changes are consistent with the HBCC algorithm principle.

According to the above analysis, when the network is congested, the HBCC algorithm improves the average saturation throughput and packet loss ratio performance of the network without substantially increasing the delay. In order to verify that the algorithm has good practicability and robustness, this paper proceeds to perform verification experiments in a relatively complex network environment with convergent and stochastic topology.

### 4.3. Aggregation Network Topology

#### 4.3.1. Parameter Settings

The aggregation network topology is shown in Figure 12. The related parameter settings are consistent with the simple tree topology, as shown in Table 2. After 10 seconds from the start of the simulation time, the transmitting node starts to generate CBR packets, which are forwarded by the intermediate node and transmitted to the sink node0. Node7, node9, and node 11 generate packets at a fixed rate of 10 packets/s, gradually changing the packet transmission rate of node8, node 10, and node12 (from 10 to 80 packets/s) to make the network change from no congestion to gradual congestion. Network performance indicators are tested under the DCF, HBCC, HSCC, and HRCC congestion control algorithms.

#### 4.3.2. Simulation Results and Algorithm Effectiveness Analysis

In the network topology of Figure 12, the average saturation throughput, buffer overflow packet loss ratio, and average end-to-end delay are shown with load variation curves under the DCF, HBCC, HSCC, and HRCC algorithms, respectively. It can be seen from Figure 13 that the average saturation throughput and buffer overflow packet loss ratio performance under the HBCC algorithm are greatly improved.

From Figure 13b, we can see that the packet loss ratio of HRCC has decreased, but its saturation throughput has not improved. This is because HRCC reduces the chance of the previous hop node accessing the channel after detecting congestion at the node. Although the packet loss ratio is reduced to some extent, the frequency at which the receiving node receives data, that is, the average saturation throughput, is also suppressed. The test indexes of the HBCC algorithm and the HSCC algorithm are close to each other, but the HBCC algorithm is relatively stable. The congestion of nodes in the actual network is accidental, so the HBCC algorithm combining the HSCC and HRCC algorithms has better congestion processing capability in the actual network.

### 4.4. Random Network Topology

#### 4.4.1. Parameter Settings

The two random network topologies analyzed in this section are shown in Figure 14. The parameter settings are shown in Table 5. The generated nodes are randomly distributed in the simulation area, and 10 transmitting nodes and 10 receiving nodes are randomly selected. The sending node starts sending data at any time. By changing the packet transmission rate, the network is changed from no congestion to congestion, and the network performance is observed under the four algorithms DCF, HBCC, HSCC and HRCC.

#### 4.4.2. Simulation Results and Algorithm Effectiveness Analysis

Figure 15, Figure 16 and Figure 17 are the graphs of average saturation throughput, buffer overflow packet loss ratio, and average end-to-end delay with load for the four algorithms in the two random network topologies shown in Figure 14. It can be seen from the figure that the average saturation throughput and buffer overflow packet loss ratio performance under the HBCC algorithm are greatly improved under the condition that the average end-to-end delay is almost unchanged. When the load increases gradually, the performance of the HBCC algorithm is better, that is, the effect of congestion alleviation is better. In the case of random network topology, the HRCC algorithm hardly works, while the HSCC algorithm and the HBCC algorithm almost coincide with the performance curve, but the performance curve of HBCC is relatively stable.

Compared with fixed topology, the effect of the HBCC algorithm in a random topology environment is slightly reduced, because all kinds of random topology situations are generated randomly. There are many sending nodes in a random network topology, which results in collisions between nodes competing for channels. The environment of the network is relatively vicious, but the HBCC algorithm still has the effect of alleviating network congestion. As can be seen from the figure, the performance curves of the HBCC algorithm and the HSCC algorithm almost coincide, and the HRCC algorithm has little effect on network congestion. In a random topology environment, with the increase of load, network congestion occurs. At this time, the number of congested nodes is greater and the distribution is dense, so network congestion cannot be greatly alleviated. Under the HRCC algorithm, when a node is congested, the transmission rate of the previous hop node is suppressed, and at this time, the previous hop node is also congested. The HRCC algorithm aggravates the congestion of the previous hop node, and the packet loss increases. This is only the transfer of packet loss for the entire network, so the congestion is not alleviated. Therefore, the performance curve of the HBCC algorithm combined with the HRCC and HSCC algorithms is almost the same as the HSCC congestion mitigation.

It can be seen from the above experiments that the HBCC algorithm has alleviated the congestion of the network under different network environments. Furthermore, when the congestion nodes in the network are relatively dispersed, the performance of the algorithm is better. In the actual ad-hoc network, the congestion is caused by a small number of nodes. Therefore, the HBCC algorithm can effectively alleviate the congestion of the network.

## 5. Conclusions

This paper first analyzed the two problems in the existing congestion control protocols and then proposed a competition-based hop-by-hop bidirectional congestion control algorithm—HBCC. The algorithm divides the congestion of two nodes in one hop into four categories, namely, 0–0, 0–1, 1–0, 1–1, and adopts corresponding congestion control strategies for four different congestion conditions. Under a simple tree topology, an aggregation network topology and a random network topology, the HBCC algorithm is simulated and compared with the DCF algorithm, HRCC algorithm, HSCC algorithm and CA-MAC algorithm. The HRCC algorithm and HSCC algorithm are unidirectional congestion control methods. CA-MAC is a congestion control method based on receiving, and its effect is similar to HRCC. Compared with the two-way HBCC algorithm, the effect of congestion mitigation has been greatly improved. The simulation results show that the HBCC algorithm greatly improves the average saturation throughput, greatly reduces the buffer overflow rate, effectively reduces the queue length of the congested nodes, and alleviates network congestion under different network environments.

## Figures and Tables

**Figure 1 sensors-19-03484-f001:**
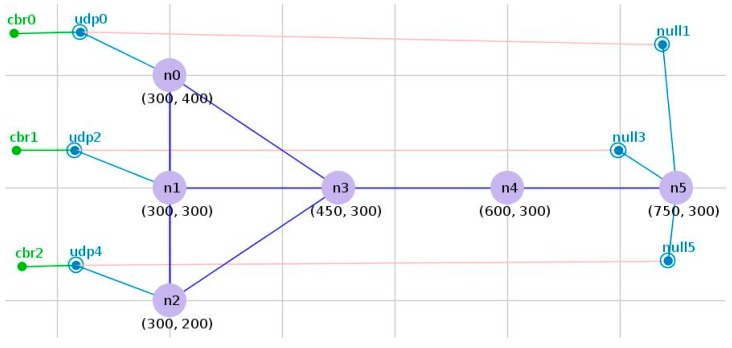
Simple tree topology.

**Figure 2 sensors-19-03484-f002:**
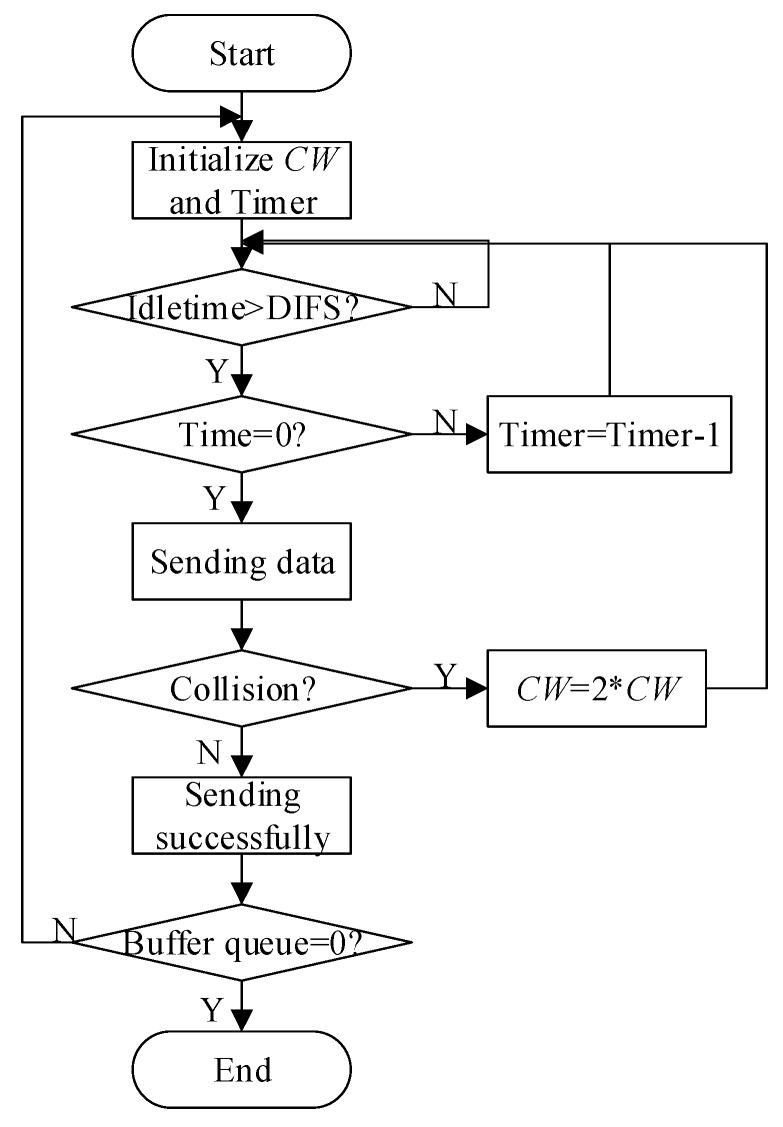
Data transmission chart.

**Figure 3 sensors-19-03484-f003:**
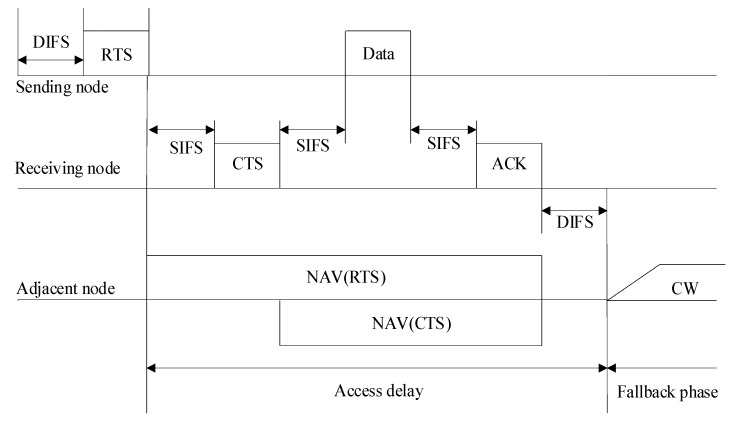
Data transmission step after the node accesses the channel.

**Figure 4 sensors-19-03484-f004:**
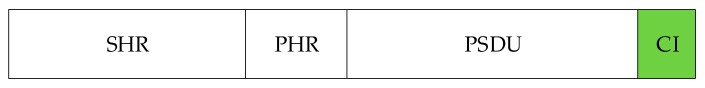
Modified ACK frame format.

**Figure 5 sensors-19-03484-f005:**
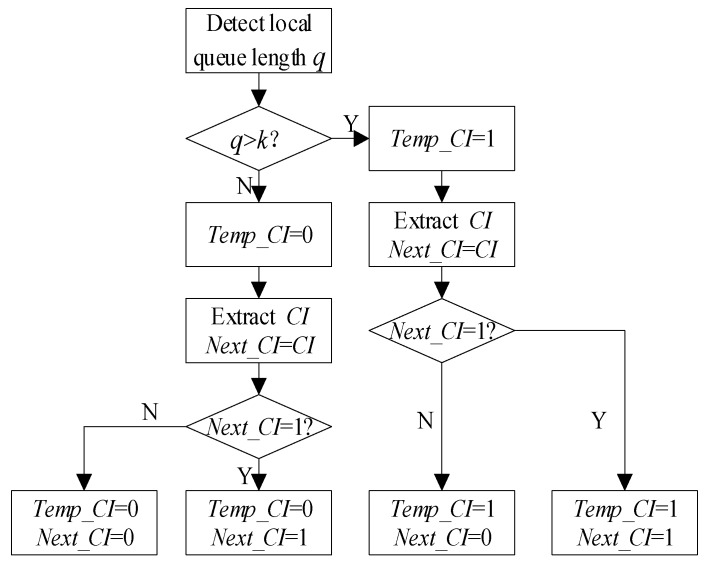
Classification of congestion conditions.

**Figure 6 sensors-19-03484-f006:**
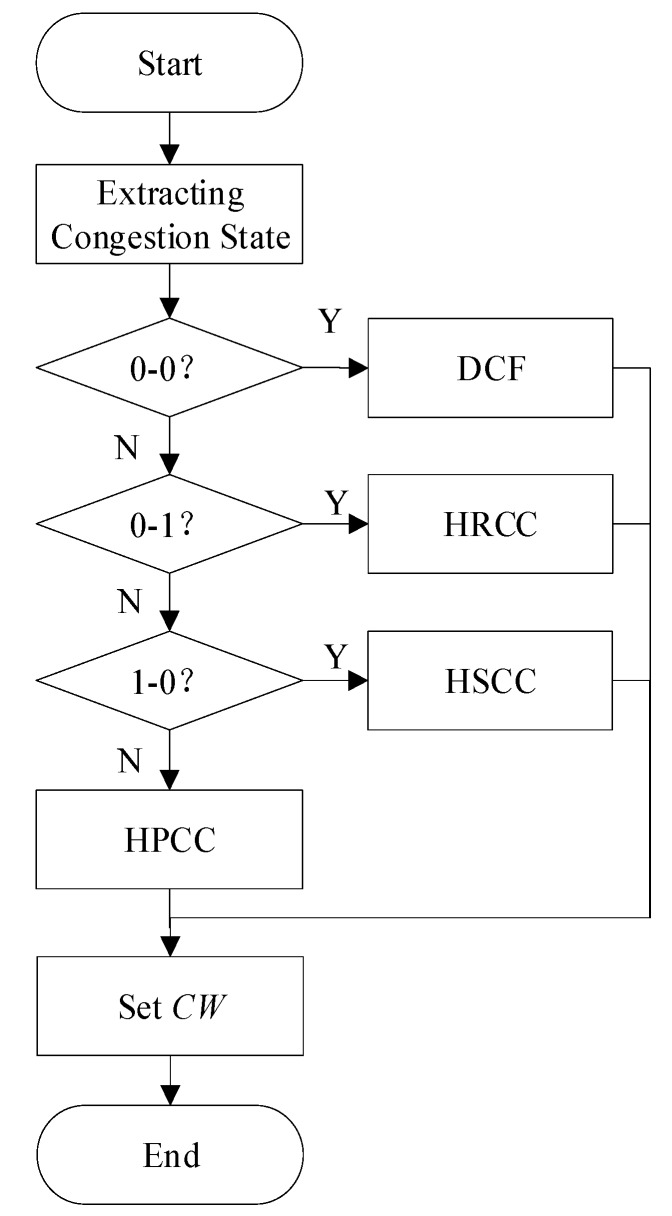
Contention window adjustment strategies of HBCC.

**Figure 7 sensors-19-03484-f007:**
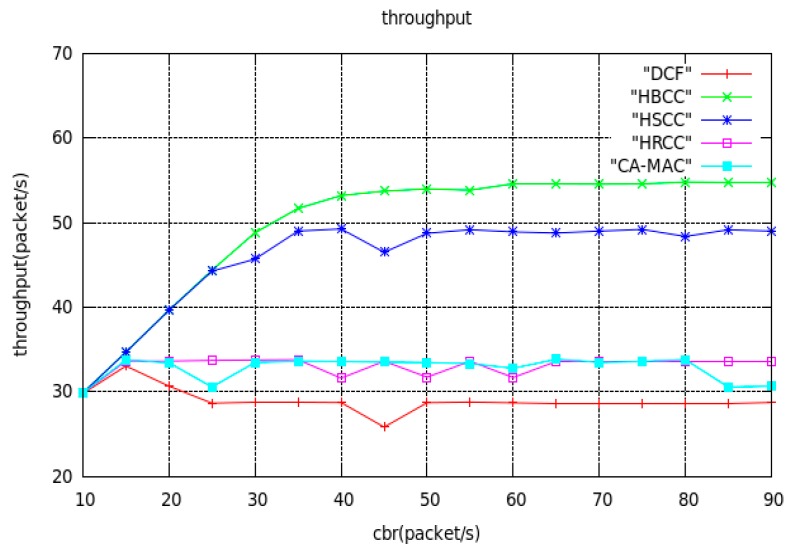
Curves of average saturation throughput with network load.

**Figure 8 sensors-19-03484-f008:**
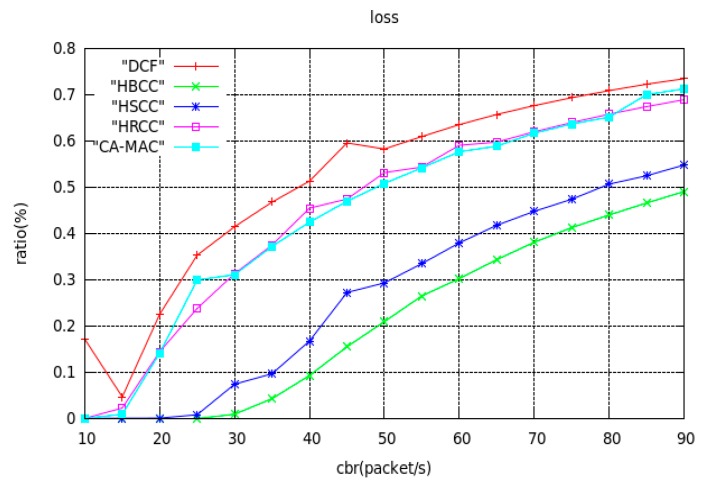
Curves of buffer overflow loss ratio with network load.

**Figure 9 sensors-19-03484-f009:**
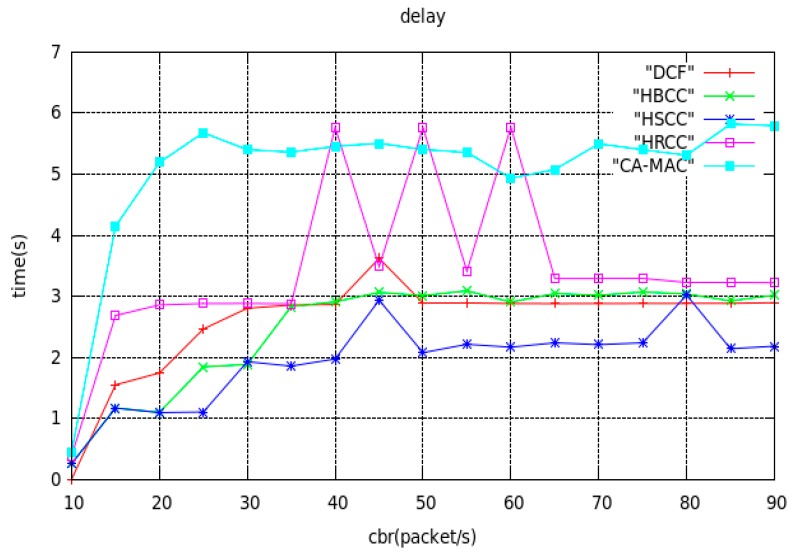
Curves of average end-to-end delay with network load.

**Figure 10 sensors-19-03484-f010:**
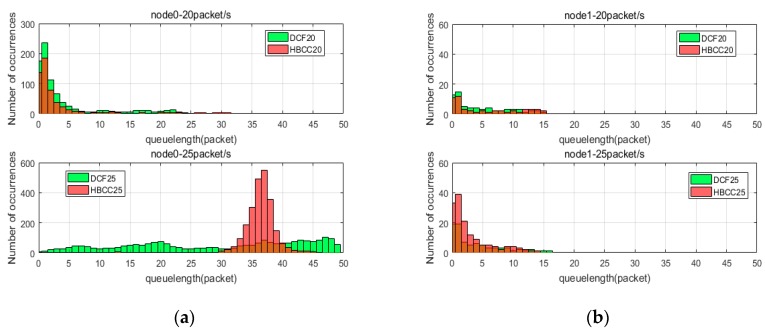
Buffer queue distribution of nodes when loads are 20 and 25 packets/s. (**a**) Node0; (**b**) Node1; (**c**) Node3; (**d**) Node4.

**Figure 11 sensors-19-03484-f011:**
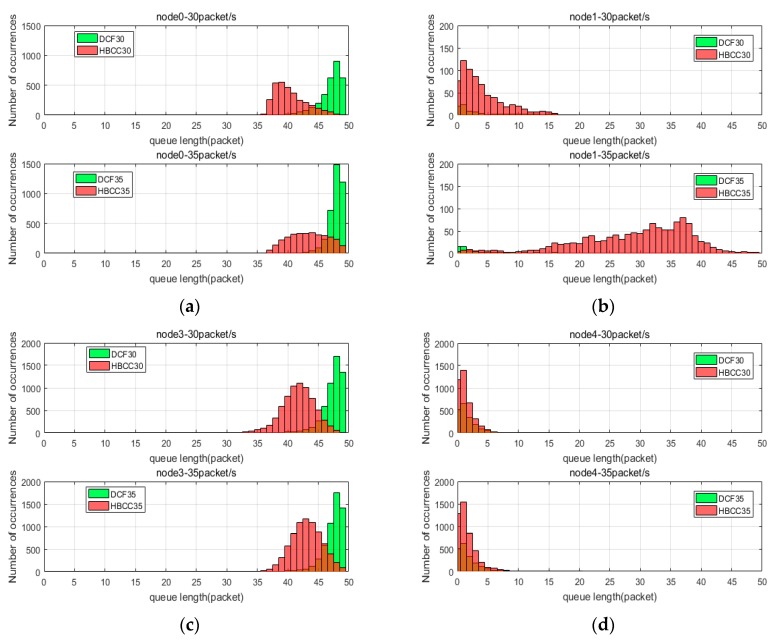
Node buffer queue distribution when loads are 30 and 35 packets/s. (**a**) Node0; (**b**) Node1; (**c**) Node3; (**d**) Node4.

**Figure 12 sensors-19-03484-f012:**
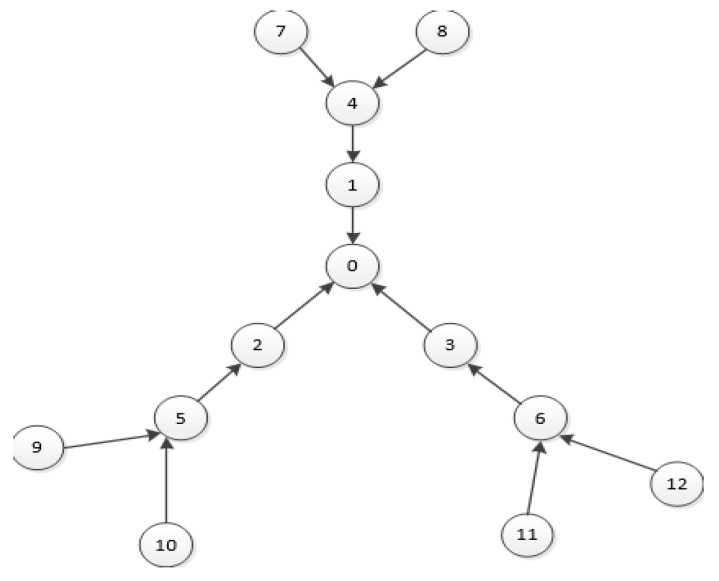
Aggregation network topology.

**Figure 13 sensors-19-03484-f013:**
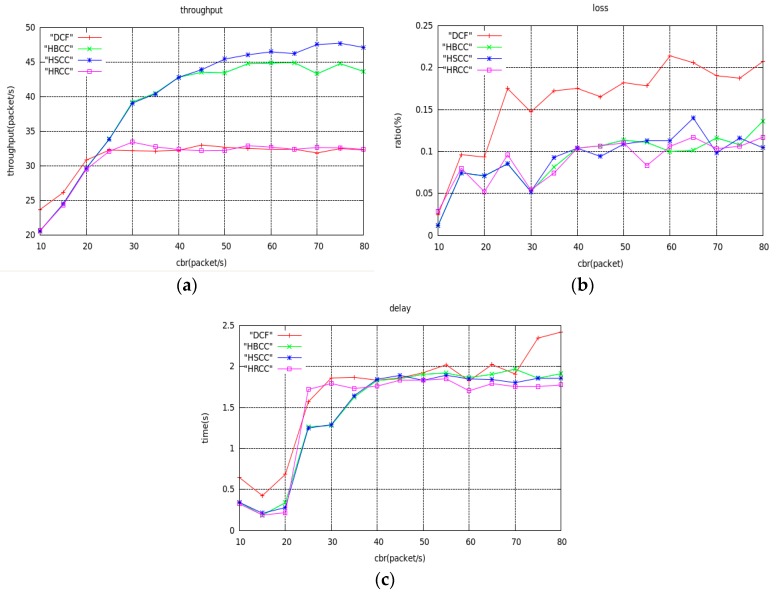
Performance simulation of aggregation topology networks. (**a**) The curves of average saturation throughput; (**b**) The curves of buffer overflow loss ratio; (**c**) The curves of average end-to-end delay.

**Figure 14 sensors-19-03484-f014:**
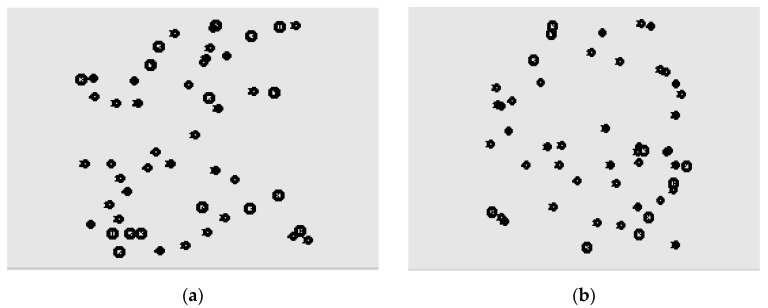
Two random network topologies. (**a**) Random network topology 1; (**b**) Random network topology 2.

**Figure 15 sensors-19-03484-f015:**
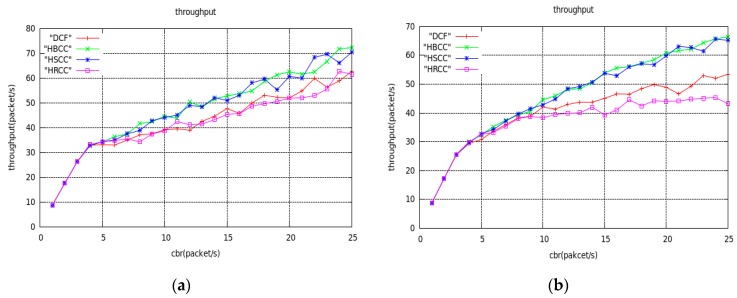
Average throughput curves of random network topology. (**a**) The topology of Figure 14a; (**b**) The topology of Figure 14b.

**Figure 16 sensors-19-03484-f016:**
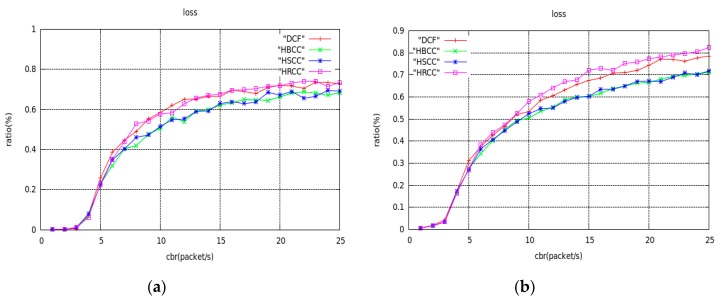
Buffer overflow packet loss ratio curves of random network topology. (**a**) The topology of Figure 14a; (**b**) The topology of Figure 14b.

**Figure 17 sensors-19-03484-f017:**
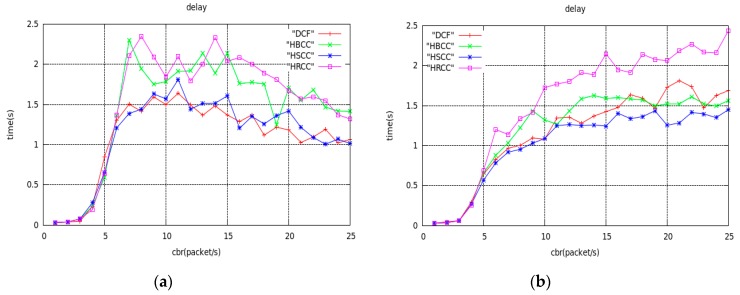
Average end-to-end delay curves of random network topology. (**a**) The topology of Figure 14a; (**b**) The topology of Figure 14b.

**Table 1 sensors-19-03484-t001:** Network Parameter.

Name	Settings
bandwidth	11 Mb
CPThresh	10.0
CSThresh	501187 × 10^−12^
RXThresh	5.82587 × 10^−9^
basicRate	1 Mb
Channel type	Wireless channel
Routing protocol	AODV
Link layer type	LL
CBR packetsize	1024 B
UDP PacketSize	1500 B
Max packet in ifq	50 packet
Mac type	IEEE 802.11
Interface queue type	Queue/DropTail/PriQueue

**Table 2 sensors-19-03484-t002:** Parameter Settings.

Name	Settings
Environment	Win7 + VMware + Ubuntu10.04 + NS − 2.35
Queue management	Droptail
Routing Protocol	AODV
Node communication radius	250 m
Packet type	CBR
Packet size	1024 B
Maximum buffer queue length	50 packet
Simulation duration	250 s
CW_min_	31
CW_max_	1023

**Table 3 sensors-19-03484-t003:** Queue status of nodes.

Load Packet/s	Algorithm	Average Queue Length of Nodes (Packet)					Nodes with Full Queue	Packet Loss Ratio
		Node0	Node1	Node2	Node3	Node4		
20	DCF	1.53	0.16	0.22	44.28	0.94	Node3	22.5%
	HBCC	0.98	0.11	0.20	35.45	0.57	No	No
25	DCF	26.85	0.12	0.14	45.02	0.93	Node 0 Node 3	35.3%
	HBCC	33.73	0.27	0.40	37.69	0.69	No	No
30	DCF	43.21	0.13	0.13	44.38	0.94	Node 0 Node 3	41.4%
	HBCC	37.67	2.06	1.25	39.60	0.99	Node 3	0.9%
35	DCF	44.67	0.10	0.31	44.55	1.08	Node 0 Node 3	46.8%
	HBCC	40.55	26.76	23.98	40.99	1.21	Node 0 Node 3	4.3%

**Table 4 sensors-19-03484-t004:** Adjustment effect of the HBCC algorithm.

Load Packet/s	Node Combination	Congestion Conditions	Strategies of CW	Average Queue Length Change Compared with DCF			
				Node0	Node1	Node3	Node4
20	0–3	0–0, 0–1	DCF, HRCC	similar		decline	
	1–3	0–0, 0–1	DCF, HRCC		similar	decline	
	3–4	0–0, 1–0	DCF, HSCC			decline	similar
25	0–3	0–1, 1–1	HRCC, HPCC	rise		decline	
	1–3	0–0, 0–1	DCF, HRCC		rise	decline	
	3–4	0–0, 1–0	DCF, HSCC			decline	similar
30	0–3	0–1, 1–1	HRCC, HPCC	decline		decline	
	1–3	0–1	HRCC		rise	decline	
	3–4	1–0	HSCC			decline	rise
35	0–3	1–1	HPCC	decline		decline	
	1–3	0–1, 1–1	HRCC, HPCC		rise	decline	
	3–4	1–0	HSCC			decline	rise

**Table 5 sensors-19-03484-t005:** Parameter Settings.

Name	Settings
Environment	Win7 + VMware + Ubuntu10.04 + NS − 2.35
Queue management	Droptail
Routing Protocol	AODV
Node communication radius	250 m
Packet type	CBR
Packet size	1024 B
Maximum buffer queue length	50 packet
Simulation duration	300 s
Simulation area	800 m × 800 m
Number of nodes	50
Number of communicable nodes	50
Sending node	10
Receiving node	10
CW_min_	31
CW_max_	1023

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
