# Peer review of "A Contention-Based Hop-By-Hop Bidirectional Congestion Control Algorithm for Ad-Hoc Networks"

_sensors, 2019, doi:10.3390/s19163484_

Round 1

Reviewer 1 Report

This work propose a new solution to solve the congestion issues in a supposed WSN. However, this study does not take into account the true context of a WSN network. For example, Table 1 refers to packets of 1024 byte  size and 50 packets queue size per node, which could be attributed to a WLAN like WIFI network rather than any WSN. In addition, reference is often made to the DCF (IEEE 802.11 Wifi context), rather than IEEE 802.15.4 CSMA-CA, which is the referral standard in the field of sensor networks!  

The bite rates in equation (8) are also exaggerated, a 10Mbps for a WSN is too high !!

I propose to use to the following work to take into account the WSN context:

Imen Bouazzi, Jamila Bhar, Mohamed Atri, "Analysis of the IEEE 802.15.4 MAC Parameters to Achieve Lower Packet Loss Rates," Computer Science, Volume 73, 2015, Pages 443-451,

In line 161, the authors propose to use a value 75% for k parameter. How is this value justified? It is part of the proposed solution parameters, if it changes the proposed method will be impacted.  This value should be discussed further to determine the most suitable K parameter values? for example, the topology of figure 1 can be used to determine the optimum value of K.

The same comment can be applied to equation 8 for values :  0.1 and 0.25.

Figures 10 and 11 are unclear, the intersection area is not visible between red and green. Ideally the intersection area should be with another color. For example,  use yellow for DCF, blue for HBCC and green for the intersection areas of the distributions. Also, the X and Y axis labels of these figures are not readable. In Figure 3, replace "note" with "node"

For the bibliography, this one needs some updating to take into account recent work in the field of congestion in the WSN, for example:

Ghaffari, Ali. "Congestion control mechanisms in wireless sensor networks: A survey." Journal of Network and Computer Applications 52 (2015): 101-115.

Jan, Mian Ahmad, et al. "A comprehensive analysis of congestion control protocols in wireless sensor networks." Mobile networks and applications 23.3 (2018): 456-468.

Rajesh, M. "Control Plan Transmits to Congestion Control for AdHoc Networks." Universal Journal of Management and Information Technology (UJMIT) 1.1 (2016): 8-11.

Kaur, Jasleen, Rubal Grewal, and Kamaljit Singh Saini. "A survey on recent congestion control schemes in wireless sensor network." 2015 IEEE International Advance Computing Conference (IACC). IEEE, 2015.

Jan, Mian Ahmad, et al. "A comprehensive analysis of congestion control protocols in wireless sensor networks." Mobile networks and applications 23.3 (2018): 456-468.

Shah, Syed Afsar, Babar Nazir, and Imran Ali Khan. "Congestion control algorithms in wireless sensor networks: Trends and Opportunities." Journal of King Saud University-Computer and Information Sciences 29.3 (2017): 236-245.

Ma, Chuang. "A congestion control protocol for wireless sensor networks." International Conference on Computational Social Networks. Springer, Cham, 2018.

Reviewer 2 Report

In this paper, authors propose a contention-based hop-by-hop bidirectional congestion control algorithm. The paper is well structured and addresses a recurrent problem that happens in WSN. Authors review the main algorithms to do the same and evaluate it in order to promote their proposal. The results are very interesting. Authors promote a comprehensive comparison between the state of the art and the new proposal. However, those results show a big improvement concerning the other approaches. I tried to understand the reason from the theorical part and in my opinion some parts/description are missing. The algorithm present in the form of pseudo-code are not well described. The definition of some variables is missing. Moreover, theorical parts are not well described and supported by other works (in my opinion some references are missing to support some concepts).

Reviewer 3 Report

This paper presents a bidirectional congestion control algorithm for wireless sensor networks. The paper is in general easy to follow, however there are some major issues that do not allow its acceptance at this phase:

1) There is no system model. The authors briefly provide a problem statement in a toy example (topology with 5 nodes) and then they introduce their algorithm. A section where the network topology and the parameters are presented is completely missing, so it is not possible to evaluate the applicability and scalability of the proposed solution.

2) It is not clear whether HRCC and HSCC are existing approaches in the literature or they are introduced in this paper. The authors use also these acronyms in the abstract without any explanation. If these solutions exist, then it seems that the paper has no contribution as it simply combines these techniques. If these techniques are proposed in this work, then the authors should compare their method with other existing approaches.

3) There is no theoretical analysis to validate the simulation results for the proposed method.

4) Although the paper is in general easy to follow, its presentation needs significant imporvement as the writing is informal in many cases, while there are several typos and grammar errors (e.g., "sending note", etc.)

Round 2

Reviewer 1 Report

Thank you for making some changes, but I still see that there is a conflict between the main work context of the paper that is more about the wireless ad-hoc network than addressing a wireless sensor network issues. The proposed work is useful for any ad-hoc network using DCF as a medium access control protocol, why the paper is limited to only wireless sensors uses case? 

It would be more accurate to rectify by targeting a wireless ad hoc network, and highlighting the WSN uses case. 

Reviewer 3 Report

In this revised version, the authors tried to improve the paper. However, there are still several issues that have not been addressed:

1) As in the previous version, the system model is missing. System model should explain all the aspects of the network in a generalized way. Fig. 1 defines a toy example of 5 nodes, while no parameters are defined.

2) There are still typos and grammar errors, while the writing in many cases is confusing (e.g. the authors use the past tense to explain the contributions of their work in the introduction, etc.)

3) It is not clear which is the bidirectionality of the proposed algorithm. In all examples in the paper, the flows seem to be unidirectional, i.e., towards one direction. It is very hard for the reader to follow the concepts.
